# CompGS: Unleashing 2D Compositionality for Compositional Text-to-3D via Dynamically Optimizing 3D Gaussians

## ABSTRACT

Recent breakthroughs in text-guided image generation have significantly advanced the field of 3D generation. While generating a single high-quality 3D object is now feasible, generating multiple objects with reasonable interactions within a 3D space, *a.k.a.* compositional 3D generation, presents substantial challenges. This paper introduces CompGS, a novel generative framework that employs 3D Gaussian Splatting (GS) for efficient, compositional text-to-3D content generation. To achieve this goal, two core designs are proposed: (1) *3D Gaussians Initialization with 2D compositionality*: We transfer the well-established 2D compositionality to initialize the Gaussian parameters on an entity-by-entity basis, ensuring both consistent 3D priors for each entity and reasonable interactions among multiple entities; (2) *Dynamic Optimization*: We propose a dynamic strategy to optimize 3D Gaussians using Score Distillation Sampling (SDS) loss. CompGS first automatically decomposes 3D Gaussians into distinct entity parts, enabling optimization at both the entity and composition levels. Additionally, CompGS optimizes across objects of varying scales by dynamically adjusting the spatial parameters of each entity, enhancing the generation of fine-grained details, particularly in smaller entities. Qualitative comparisons and quantitative evaluations on $T^3$Bench demonstrate the effectiveness of CompGS in generating compositional 3D objects with superior image quality and semantic alignment over existing methods. CompGS can also be easily extended to controllable 3D editing, facilitating complex scene generation. We hope CompGS will provide new insights to the compositional 3D generation. Codes will be released to the research community.

## 1 INTRODUCTION

3D content creation is essential to the modern media industry, yet it has traditionally been labour-intensive and necessitated professional expertise. Typically, designing a single 3D object takes several hours for an experienced designer, and creating complex scenes with multiple 3D objects (as shown in Fig. 1) requires even more effort. Inspired by the recent success of diffusion models in the text-to-image generation Ho et al. (2020); Song et al. (2020a); Rombach et al. (2022a); Podell et al. (2023), much research has focused on exploring 2D diffusion models for text-to-3D generation. Previous work can be divided into two main methodologies: (1) *Feed-forward generation* Li et al. (2023); Hong et al. (2023); Xu et al. (2024), which entails training generalizable diffusion models on 3D data; and (2) *Optimization-based generation* Poole et al. (2022); Lin et al. (2023); Metzer et al. (2023); Chen et al. (2023b); Wang et al. (2023a; 2024), which utilizes the pretrained 2D diffusion guidance to optimize 3D representations via Score Distillation Sampling (SDS) Poole et al. (2022).

While existing work has demonstrated the feasibility of generating single 3D objects, they often struggle to produce compositional 3D content with multiple objects and complex interactions. For example, (1) Feed-forward generation methods struggle in generalizing to complex textual descriptions since the amount of 3D training data is extremely limited Deitke et al. (2023; 2024), and most of the data contains only one object; (2) Optimization-based generation methods face significant challenges with current 2D diffusion guidance when optimizing compositional 3D objects. Typically, using 2D diffusion guidance to optimize a single object is feasible, as it's easy to incorporate various attributes into a single object. However, in compositional 3D generation, 2D guidance struggles to accurately

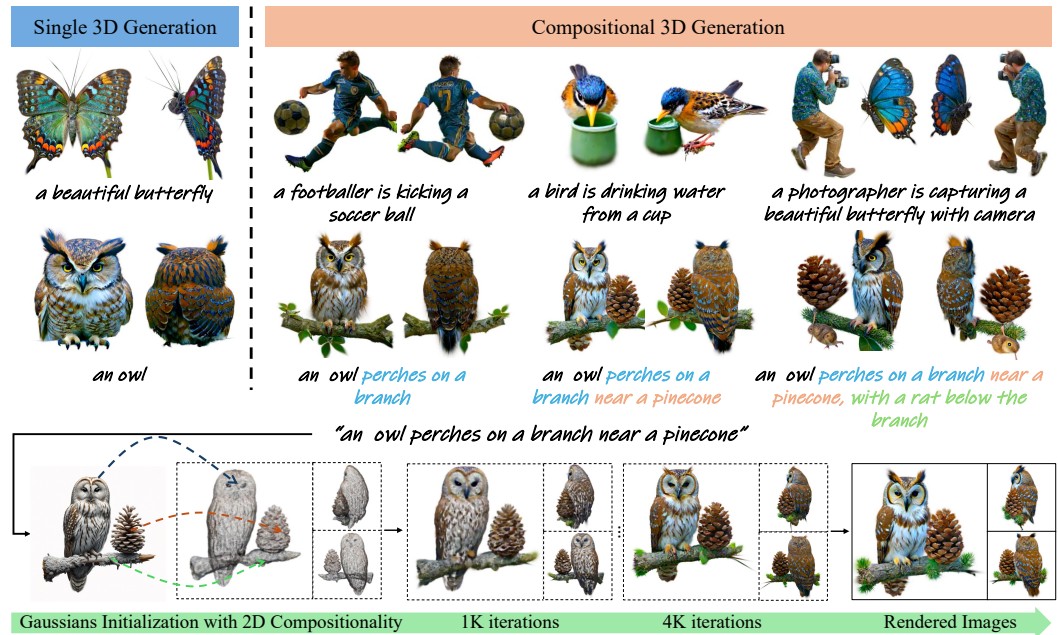

Figure 1: **Illustration of compositional 3D Generation and COMPGS.** All the contents are generated by COMPGS. *Top row:* COMPGS is capable of generating either a single object (e.g., a butterfly) or generating compositional objects with reasonable interactions (e.g., the rightmost figure in the top row). *Middle row:* Beyond text-to-3D generation, COMPGS can be easily extend to 3D editing by progressively adding objects. The colored texts (e.g., *'a branch', 'a pinecone', 'a rat'* in the rightmost figure) denote the added part compared to its previous asset. *Bottom row:* COMPGS achieves compositional text-to-3D by transferring 2D compositionality to initialize 3D Gaussians. COMPGS is further trained with dynamic SDS optimization to produce plausible results.

compose multiple objects with different attributes and relationships into a coherent scene Huang et al. (2023). For example, given the prompt *'a blue bench on the left of a green car,'* distinguishing different attributes within the implicit 2D diffusion priors is challenging, resulting in misaligning attributes to different objects or generating incorrect spatial relationships. Thus, optimization-based compositional generation with standard 2D guidance often causes problems like 3D inconsistencies, multi-faced objects, semantic drift, etc Shi et al. (2023); He et al. (2023).

In this work, we propose COMPGS, a generative system based on 3D Gaussian Splatting (GS) Kerbl et al. (2023) for compositional text-to-3D generation. To achieve this, we introduce two core designs:

**3D Gaussians Initialization with 2D Compositionality** Unlike the implicit representation, i.e., NeRF Mildenhall et al. (2021), COMPGS uses 3D Gaussians as the representation, which helps to achieve feasible parameter initialization with a coarse 3D shape Jun & Nichol (2023); Yi et al. (2023). As shown in the bottom row of Fig. 1, we first apply a text-to-image model Betker et al. (2023); Ramesh et al. (2021); Chen et al. (2023a) to generate an image that accurately captures the compositionality of multiple objects. Then the image is segmented into different sub-objects (*a.k.a.* entities) according to the entity information in the given prompt. These segmented entities are processed through an image-to-3D model Hong et al. (2023); Tochilkin et al. (2024) to obtain coarse 3D shapes, which are used to roughly initialize the Gaussian parameters in 3D space, thereby transferring the 2D compositionality to 3D representations.

**Dynamic Optimization**. Current 2D guidance Poole et al. (2022); Lin et al. (2023); Metzer et al. (2023) struggles with optimizing multiple 3D objects simultaneously in a scene, often leading to 3D inconsistencies and semantic shifts. To address these issues, we introduce a dynamic optimization strategy based on Score Distillation Sampling (SDS) loss, consisting of two key components: (1) COMPGS dynamically divides the training process to optimize different parts of 3D Gaussians. Specifically, it alternates between optimizing a single object (entity-level optimization) and the entire scene (composition-level optimization). This is achieved by labeling and filtering the Gaussian

parameters for inference, and updating them through masking gradients. (2) COMPGS dynamically trains the entity-level Gaussians within a normalized 3D space, crucial for compositional 3D generation where object sizes vary. This is particularly challenging when dealing with very small objects, as optimizing Gaussian parameters in a limited 3D space may not adequately capture detailed textures. To mitigate this, we first scale the subspace of each entity to a predefined, standardized volume. Following each training iteration within these standardized volumes, we rescale the Gaussian parameters back to their original sizes. This method dynamically maintains volume consistency throughout the training process, thereby facilitating the capture of fine details within smaller objects.

With these designs, COMPGS is capable of generating high-quality compositional 3D objects (as shown in the first row of Fig. 1) and progressive 3D editing (as shown in the second row of Fig. 1). We demonstrate the effectiveness of COMPGS both qualitatively and quantitatively. Qualitative comparisons through our user study indicate that COMPGS offers superior image quality and semantic alignment compared to existing models (e.g., Fantasia3D Chen et al. (2023b), ProlificDreamer Wang et al. (2024), VP3D Chen et al. (2024c), etc.). Besides, COMPGS's performance on $T^3$Bench He et al. (2023) quantitatively highlights its advantages in both semantic control and high-fidelity generation. We hope COMPGS can provide valuable insights into the research of compositional 3D generation.

The contributions of this work are as follows: (1) We introduce COMPGS, a user-friendly generative system framework for compositional 3D generation based on 3D Gaussians, which produces high-quality multiple 3D objects with complex interactions. (2) COMPGS transfers 2D compositionality to facilitate composed 3D generation and incorporates dynamic SDS optimization to address challenges in maintaining 3D consistency, generating plausible shapes and textures, and formulating reasonable object interactions. (3) COMPGS demonstrates superior performance compared to previous methods in compositional text-to-3D generation, both quantitatively and qualitatively, and can be easily extended to progressive 3D editing.

## 2 RELATED WORK

**Multi-modality 3D Generation** Multi-modality 3D generation can basically categorized into the feed-forward system and optimization-based system. The former one is trained end-to-end on multi-view dataset Chang et al. (2015); Deitke et al. (2023) for zero-shot generation, typically based on 3D representations, including NeRF Hong et al. (2023); Tochilkin et al. (2024), 3D Gaussians Xu et al. (2024); Tang et al. (2024), tri-planes  Shue et al. (2023); Wang et al. (2023b) and feature grids Karnewar et al. (2023). Despite the excellent performance achieved, the single-object training data Deitke et al. (2023); Luo et al. (2024); Deitke et al. (2024); Chang et al. (2015) limits their generative abilities in scenarios involving multiple objects. Optimization-based 3D systems lift 2D diffusion priors Rombach et al. (2022a;b); Podell et al. (2023); Chen et al. (2023a; 2024a) for 3D generation, and typically train 3D representations on a prompt-by-prompt basis. For example, DreamFusion Poole et al. (2022) introduces Score Distillation Sampling (SDS) loss to transfer 2D diffusion models into the 3D domain. Magic3D Qian et al. (2023) employs a coarse-to-fine scheme to improve both efficiency and effectiveness, while Fantasia3D Chen et al. (2023b) decouples the modelling of geometry and appearance. Despite the advancements of these methods addressing various challenges, they also usually struggle with compositional 3D generation. $T^3$bench's examination of ten prominent optimization-based models Poole et al. (2022); Lin et al. (2023); Metzer et al. (2023); Wang et al. (2024; 2023a); Chen et al. (2023b); Shi et al. (2023); Yi et al. (2023) revealed frequent issues in compositional generation. The implicit 2D diffusion guidance used in these methods Poole et al. (2022); Shi et al. (2023); Wang et al. (2024); Chen et al. (2023b) often fails to maintain 3D consistency across different views, leading to significant discrepancies in compositional 3D scenes.

**Compositional Generation** Compositional generation involves creating content involving multiple objects with logical interactions. These interactions include, but are not limited to, concept relations Liu et al. (2022a), attribute association with colors Chefer et al. (2023); Feng et al. (2022); Park et al. (2021), and spatial relationships between objects Wu et al. (2023); Chen et al. (2024b). There is considerable focus on various aspects of compositional 2D generation, such as learning from human feedback Zhang et al. (2023); Lee et al. (2023); Dong et al. (2023); Yang et al. (2024), enhancing image captions Chen et al. (2023a; 2024a); Betker et al. (2023); Dai et al. (2023), designing effective networks Liu et al. (2022b), and etc. T2I-CompBench Huang et al. (2023) further proposed a comprehensive benchmark and boosted the compositionality of text-to-image models. Compared to 2D compositional generation, its 3D counterpart is under-explored due to 3D geometry and appearance

complexities. For example, Set-the-Scene Cohen-Bar et al. (2023) and Scenewiz3d Zhang et al. (2024b) propose to adopt object layouts to generate compositional scenes. However, they are mainly based on the implicit NeRF representations, so the features between objects cannot be well decoupled, resulting in a relatively blurred rendering effect. VP3D Chen et al. (2024c) takes a different strategy to employ visual features for compositional generation. However, its visual features are only used as implicit supervision, which does not adequately ensure 3D consistency or address the Janus problems Shi et al. (2023). Besides, LucidDreaming Wang et al. (2023c) requires human-annotated layout priors for local optimization, which is labour-intensive and inaccurate; while GraphDreamer Gao et al. (2024) optimizes all interactive objects from scratch, which is time-consuming and difficult to optimize. The most concurrent works, GALA3D Zhou et al. (2024), propose optimizing the spatial relationships of well-trained Gaussians for compositional generation. Though it achieves plausible spatial interactions, it is inefficient in generating complex mutual interactions among objects. More comparisons are in the appendix. We propose COMPGS to address the above challenges, emphasizing efficient optimization and the generation of complex interactions.

## 3 METHODOLOGY

In this section, we revisit 3D Gaussian Splatting Kerbl et al. (2023) and diffusion priors for 3D generation. We then provide an overview of COMPGS, which includes initializing 3D Gaussians to incorporate compositionality priors and dynamic SDS optimization to generate high-fidelity, 3D consistent objects with complex interactions. The notations used in this section are also detailed in Appendix A.1 for clarity.

### 3.1 PRELIMINARIES

**3D Gaussian Splatting (GS)** Kerbl et al. (2023) has recently revolutionized novel-view synthesis of objects/scenes by its real-time rendering. Specifically, GS represents the 3D objects/scenes by $N$ explicit anisotropic Gaussians with center positions $\mu_i$, covariances $\Sigma_i$, opacities $\alpha_i$ and colors $c_i$, where $i \in N$. The color $\mathcal{C}(\mathbf{p})$ of image pixel $\mathbf{p}$ can be calculated through point-based volume rendering Kopanas et al. (2021; 2022) by integrating the color and density of the 3D Gaussians intersected by a ray, as follows:

$$\mathcal{C}(\mathbf{p}) = \sum_{i=1}^{N} c_i \sigma_i \prod_{j=1}^{i-1} (1 - \sigma_j), \tag{1}$$

$$\sigma_i = \alpha_i \exp \left[ -\frac{1}{2} (\mathbf{p} - \hat{\boldsymbol{\mu}}_i)^\top \hat{\Sigma}_i^{-1} (\mathbf{p} - \hat{\boldsymbol{\mu}}_i) \right], \tag{2}$$

where $\hat{\boldsymbol{\mu}}_i$ and $\hat{\Sigma}_i$ denote the projected center positions and covariances of the 2D Gaussians, transformed from 3D space to the 2D camera's image plane. The image plane can be segmented into tiles during rendering for parallel processing. Unlike implicit representations, such as NeRF Mildenhall et al. (2021); Barron et al. (2021); Müller et al. (2022), GS offers two key advantages for compositional 3D generation: (1) 3D Gaussians facilitate localized rendering, allowing for the rendering of *object A* in one specific sub-space and *object B* in another, thus enabling global compositional generation on a divide-and-conquer basis; (2) 3D Gaussians enable direct parameter initialization, simplifying the integration of compositional priors at the initialization stage.

**Diffusion Priors for 3D Generation** Diffusion-based generative models (DMs) Dhariwal & Nichol (2021); Sohl-Dickstein et al. (2015); Song et al. (2020b) have been widely utilized to provide implicit priors for 3D object generation via score distillation sampling (SDS) Poole et al. (2022). Specifically, given a 3D model whose parameters are $\theta$, a differentiable rendering process $g$, the rendered images could be obtained via $\mathbf{x} = g(\theta)$. To ensure the rendered images resemble those generated by the DM $\phi$, SDS first formulates the sampled noise $\hat{\epsilon}_\phi(\mathbf{z}_t; v, t)$ with the noisy image $\mathbf{z}_t$, text embedding $v$, and noise level $t$. By comparing the difference between the added Gaussian noise $\epsilon$ and the predicted noise $\hat{\epsilon}_\phi$, SDS constructs gradients that could be back-propagated to update $\theta$ via:

$$\nabla_\theta \mathcal{L}_{\text{SDS}}(\phi, \mathbf{x} = g(\theta)) \triangleq \mathbb{E}_{t,\epsilon} \left[ w(t) (\hat{\epsilon}_\phi(\mathbf{z}_t, v, t) - \epsilon) \frac{\partial \mathbf{x}}{\partial \theta} \right], \tag{3}$$

where $w(t)$ is a weighting function, and the Classifier-free guidance (CFG) Ho & Salimans (2022) typically amplifies the text conditioning $v$.

$V=$ 'an owl perches on a branch near a pinecone'  ($v_1=$ 'an owl';  $v_2=$ 'a branch';  $v_3=$ 'a pinecone')

Figure 2: **Overall pipeline of COMPGS.** Given a compositional prompt $V$, we first use an LLM to decompose it into entity-level prompts $\{v_l\}$, guiding the segmentation of each entity from the compositional image generated by T2I models. The segmented images initialize entity-level 3D Gaussians via image-to-3D models Tochilkin et al. (2024); Hong et al. (2023). COMPGS employs a dynamic optimization strategy, alternating between composition-level optimization of $\theta$ and entity-level optimization of $\{\theta_l\}$. For entity-level optimization, COMPGS dynamically maintains volume consistency to refine the details of each objects, particularly the small one.

## 3.2   COMPGS

We propose COMPGS for efficiently transferring the 2D compositionality to facilitate compositional 3D generation with 3D Gaussians. The overall framework of COMPGS is shown in Fig. 2. We first generate a well-composed image from the given complex prompt. After extracting the sub-object (also named entity) information within the prompt by LLM, we utilize the entity prompt to segment the composed image into different parts. Each part will be adopted to initialize a specific space of Gaussians. During the training stage, we propose a dynamic SDS optimization strategy. This strategy first automatically decomposes the training to optimize either entity-level Gaussians or composition-level Gaussians. Then, it employs a volume-adaptive strategy to dynamically optimize varying-sized entities within a consistent 3D space. We detail the initialization process and dynamic SDS optimization in the following sections.

### 3.2.1   3D GAUSSIANS INITIALIZATION WITH 2D COMPOSITIONALITY

Unlike the implicit NeRF Mildenhall et al. (2021) representations, explicit 3D Gaussians can be easily initialized with 3D shapes and colors, which facilitates introducing rough 3D priors to ensure 3D consistency Yi et al. (2023). Although existing text-to-3D or image-to-3D models can generate a single 3D object, they struggle with compositional 3D generation as mentioned in Sec. 2, which hinders the creation of compositional 3D priors needed for initialization. Considering this, we propose initializing the compositional Gaussians on an entity-by-entity basis.

As shown in Fig. 2 (left), given a complex prompt $V$, (e.g., *'an owl perches on a branch near a pinecone'*), we first adopt a 2D diffusion model Betker et al. (2023) to generate a composed image $I$ that faithfully captures the compositionality of multiple objects. We extract each entity information in prompt $V$ by prompting LLM to obtain $L$ different entity-level prompts (i.e., $L = 3$ for $v_1$ *'an owl'* , $v_2$ *'a branch'* , and $v_3$ *'a pinecone'* ). These prompts are adopted in a text-guided segmentation model Kirillov et al. (2023) to decompose image $I$ into various parts $\{I_l\}$. Until now, each segmented image has only one entity, facilitating using existing image-based 3D generation models Tochilkin et al. (2024) to predict a rough triangle mesh $m_l(l \in L)$ of the corresponding 3D entity $l$. To initialize 3D Gaussians $\theta_l$, we index $N$ points from each mesh; the center positions $\mu_i^l \in \mathbb{R}^3 (i \in N, l \in L)$ are the centers of each vertex of $m_l$, and the texture colors $c_i^l \in \mathbb{R}^3 (i \in N, l \in L)$ are queried from each vertex of $m_l$. During the image-to-3D process, we did not perform any cropping or padding on the image $I_l$, ensuring that the spatial positions of each mesh $m_l$ correspond to their 2D spatial positions. Additionally, we have marked the 3D layouts of each entity asset with 3D bounding boxes $\text{bbox}_l$ for subsequent optimization. The bounding box coordinates are determined by the outer-most center positions of the entity Gaussian.

### 3.2.2 Dynamic SDS Optimization

As mentioned in Sec. 2, existing SDS methods Poole et al. (2022); Shi et al. (2023); Wang et al. (2024); Chen et al. (2023b) struggle in optimizing compositional 3D scenes due to the challenges of using implicit diffusion priors to maintain multi-objects consistency and interactions. To address this, we introduce dynamic SDS optimization to enable existing SDS losses to remain effective in compositional generation. Dynamic SDS optimization is performed through the following two procedures.

**Automatically Decomposing Optimization to Different Levels** We adopt the Decomposed Optimization (DO) strategy that divides the entire training into $L + 1$ stages, including $L$ stages for entity-level optimization to ensure 3D consistency of each $L$ entities, and one stage for composition-level optimization to refine inter-entity interactions. In each training iteration, we randomly render either (1) an entity Gaussian $\theta_l (l \in L)$ to obtain the entity image $x_l = g(\theta_l)$ ($\theta_l \in \text{bbox}_l$), or (2) the composed Gaussians $\theta$ to obtain the composed image $x = g(\theta)$. Here $g$ denotes the Gaussian Splatting rendering Kerbl et al. (2023) described in Eq. 1.

To update Gaussian parameters, we apply Eq. 3 for gradients backpropagation. Specifically, for entity-level optimization, we substitute the text embedding $v$ by $v_l$ and parameters $\theta$ by $\theta_l$ in Eq. 3 to optimize entity Gaussian. We adopt MVDream Shi et al. (2023) as $\phi$, which provides multi-view diffusion priors that can better maintain consistency in 3D entity modelling. To ensure that other entity's parameters remain unchanged, we mask the gradients that are not within the corresponding 3D bounding boxes $\text{bbox}_l$ when updating $\theta_l$. Besides, for composition-level optimization, we integrate both 2D Rombach et al. (2022a) and 3D diffusion priors Shi et al. (2023) to jointly optimize the overall Gaussian parameters $\theta$. The rationale is that 2D priors promote geometry exploration Qian et al. (2023), which enhances the generation of plausible interactions between different entities.

**Volume-adaptively Optimizing Entity-level Gaussians** Compared to the single object generation, compositional 3D generation using SDS presents additional challenges. Specifically, when objects vary in size, optimizing directly in the original 3D space often results in suboptimal generation for smaller objects, as shown the *'pinecone'* in Fig. 2. To mitigate this issue, we propose a Volume-adaptive Optimization (VAO) strategy that enhances optimization across objects of varying sizes by dynamically standardizing each entity's space to a standardized volumetric scale. This approach is especially beneficial for improving the generation of fine-grained texture details in smaller entities.

To scale the 3D bounding box of an entity Gaussian $\theta_l$ with a standardized volumetric space $\text{bbox}_{std}$, we first determine the necessary transformations for the centre positions $\mu$. Specifically, we calculate the shift parameters $\beta$, and scale parameters $\lambda$, as follows:

$$\beta = \text{Mean}(\text{bbox}_l); \quad \lambda = \text{bbox}_{std}/\text{bbox}_l, \tag{4}$$

where $\text{Mean}$ computes the center coordinates of the given bounding box, and $\text{bbox}_{std}$ is the target standardized space. Then, we zoom-in the entity Gaussian on its center positions by $\hat{\mu} = \lambda(\mu - \beta)$, where $\hat{\mu}$ denotes the transformed center positions of entity Gaussian. We substitute all $\mu$ to $\hat{\mu}$ in Eq. 1 for SDS optimization. After each training iteration, we perform zooming-back via $\mu = \hat{\mu}/\lambda + \beta$. The proposed transformation ensures dynamically conducting SDS optimization on each entity Gaussian at a standardized and consistent scale. Experiments in Sec. 4.3 demonstrate its effectiveness in generating high-quality 3D assets.

### 3.3 Progressive Editing with CompGS

The dynamic SDS optimization enables CompGS to explicitly control the generation of different parts in the 3D scene. This capability can be utilized for progressive 3D editing in compositional generation. Specifically, given well-trained 3D Gaussians $\theta$, we begin by rendering an image from the front view, denoted as $x = g(\theta)$. Unlike previous works that directly edit 3D objects Cheng et al. (2024), we adopt MagicBrush Zhang et al. (2024a) to edit the 2D rendered image $x$. The edited 2D image can then be utilized to initialize new Gaussians $\hat{\theta}$ in the original 3D space. We further train the added 3D Gaussians to finally introduce new objects into the compositional scenes. For example, as shown in Fig. 1, given well-trained 3D Gaussians (e.g., 'an owl'), we progressively edit the 2D images to add new objects, such as the *'branch'* and *'pinecone.'* We then train the corresponding 3D Gaussians via dynamic SDS optimization to incorporate these new elements into the 3D scenes.

## 4 EXPERIMENTS

### 4.1 EXPERIMENTAL SETTINGS

**Implementation Details** We implemented COMPGS using ThreeStudio Guo et al. (2023). We use DALL·E Betker et al. (2023) to generate the well-established 2D compositionality, LangSAM Medeiros (2024) to segment the entity image, and TripoSR Tochilkin et al. (2024) to generate a preliminary 3D prior (i.e., mesh) from each entity image, respectively. During the training, we employed MVDream Shi et al. (2023) to optimize entity-level Gaussians, and both MVDream Shi et al. (2023) and *stabilityai/stablediffusion-2-1-base* Rombach et al. (2022b) to optimize composition-level Gaussians. The guidance scale for diffusion models was set as 50. Timestamps were uniformly selected from 0.02 to 0.55 for the first 1,000 iterations and then adjusted to a range from 0.02 to 0.15 for subsequent iterations. We initialized the Gaussian points at $N = 10k$, and progressively increased the density to a maximum of $1000k$ points. We optimized the overall compositional scenes through 10k iterations to achieve optimal results. However, it is worth noting that we empirically found that training with 5k iterations already produces plausible 3D assets. The learning rates are detailed in Appendix A.3. We set the camera parameters, including radius, azimuth, elevation and FoV (field of view) to be the same as Shi et al. (2023). Experiments were conducted on NVIDIA A100 GPUs(40G). Our code will be released.

**Evaluation Metrics** In addition to user studies and qualitative comparisons, we utilize T$^3$Bench He et al. (2023) to render 300 prompts in order to evaluate compositional 3D generation on two criteria: (1) the visual quality of the 3D objects and (2) the alignment between the 3D objects and the input prompt. For quality evaluation, we captured multi-focal and multi-view images and sent them to text-image scoring models, CLIP Radford et al. (2021), to obtain an average quality score of the generated 3D scene. Regarding the textual alignment, we first followed Luo et al. (2024) to perform 3D captioning via BLIP Li et al. (2022) and GPT4, and then computed the recall of the original prompt within the generated caption via ROUGE-L Lin (2004).

### 4.2 PERFORMANCE COMPARISONS AND ANALYSIS

**Qualitative Evaluation** In Fig. 3, we present qualitative comparisons of compositional 3D generation. Our model, COMPGS, is compared with several open-source methods, including DreamFusion Poole et al. (2022), Magic3D Qian et al. (2023), Latent-NeRF Metzer et al. (2023), Fantasia3D Chen et al. (2023b), SJC Wang et al. (2023a), and ProlificDreamer Wang et al. (2024), as well as methods specifically designed to handle intricate text prompts, VP3D Chen et al. (2024c) in Fig. 3. The results demonstrate that COMPGS generates compositional 3D objects with superior quality, greater consistency, and more plausible interactions. DreamFusion, for example, fails to generate reasonable multi-objects from compositional prompts, revealing limitations in SDS loss for optimizing multiple objects simultaneously. Magic3D and Latent-NeRF, though guided by a 3D mesh prior, struggle with generating complex and compositional 3D geometry. Despite employing advanced SDS loss, Fantasia3D, SJC, and ProlificDreamer only improve the appearance of the content but do not fundamentally address multi-object generation. Notably, VP3D employs features from compositional images and human feedback to guide compositional 3D generation. However, it often produces unexpected artifacts (e.g., redundant dots in the top two rows of Fig. 3) or less plausible interactions (e.g., strange spatial relationships in the fourth row). This is likely due to the insufficient utilization of 2D compositionality, as VP3D adopts the implicit image features to guide optimization. In contrast, COMPGS generates high-quality and composed content that strictly aligns with the given complex prompts, demonstrating its effectiveness over others. We provide more visual comparisons with both the close-source and open-source methods (including GALA3D Zhou et al. (2024) in Fig. 9, GraphDreamer Gao et al. (2024), DreamGaussian Tang et al. (2023) in Fig. 10, etc.) in Appendix A.4 as well as the attached video.

We also demonstrate COMPGS's capabilities in generating more diverse contents in Fig. 4. COMPGS accurately generates both simple spatial relationships (shown in the first row), and more complex interactions, where objects should react to each other (shown in the second row). Besides, COMPGS is not limited to generating just two objects, it can generate multiple objects by expanding the number of entity-level Gaussians. For example, in the third row, COMPGS generates scenes with three and four entities, each with high-quality 3D shapes and textures.

*'A florist is making a bouquet with fresh flowers'*

*'An artist is painting on a blank canvas'*

*'Hot popcorn jump out from the red striped popcorn maker'*

*'A half-eaten sandwich sits next to a lukewarm thermos'*

*'Two silken, colorful scarves are knotted together'*

| DreamFusion | Magic3D | LatentNeRF | Fantasia3D | SJC | ProlificDreamer | VP3D | COMPGS (ours) |

Figure 3: **Qualitative comparisons between COMPGS and other text-to-3D models on T³Bench (multiple objects track).** Compared to others, COMPGS is better at generating highly-composed, high-quality 3D contents that strictly align with the given texts. Watch the animations by **clicking** them (Not all PDF readers support playing animations. Best viewed in Acrobat/Foxit Reader).

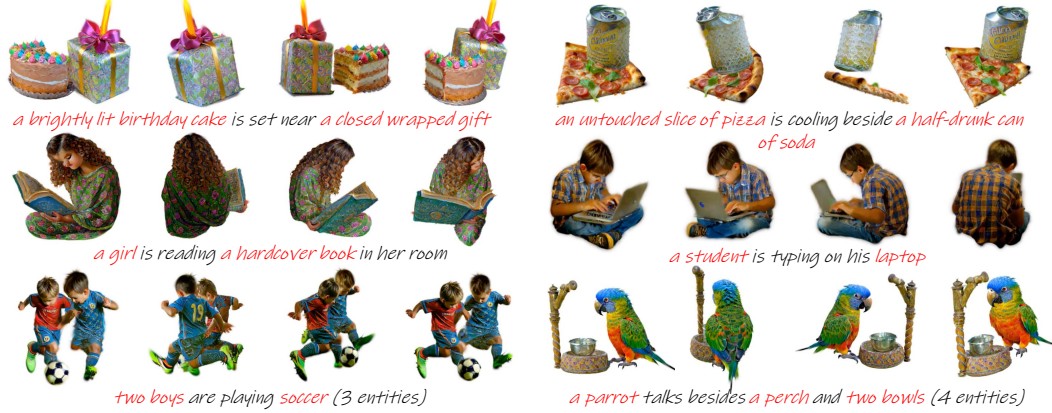

a brightly lit birthday cake is set near a closed wrapped gift

an untouched slice of pizza is cooling beside a half-drunk can of soda

a girl is reading a hardcover book in her room

a student is typing on his laptop

two boys are playing soccer (3 entities)

a parrot talks besides a perch and two bowls (4 entities)

Figure 4: **More generated samples by COMPGS.** Four views are shown. COMPGS can generate high-quality contents with reasonable interactions given two, three or more entities.

We conduct a user study for further evaluation. We randomly selected 15 prompts in T³Bench, and collected the 3D assets generated by different models. These collected 3D assets were then distributed to individuals for ranking the models based on (1) 3D visual quality and (3) text-3D alignment. We average the ranking of different models as their scores. Results in Tab. 1 show that representative optimization-based models received relatively low average scores, highlighting their limitations in compositional generation. VP3D Chen et al. (2024c) outperforms its predecessors due

to the incorporation of image features but still ranks lower than COMPGS, which further validates our effectiveness.

**Quantitative Evaluation** In Tab. 1, we benchmark the representative models on T³Bench, focusing on the multi-objects track. Results indicate COMPGS achieves superior performance in both quality and textual alignment. Compared with feedforward methods in the first block, COMPGS significantly improves the generation quality by a large margin. Though COMPGS is initialized from the 3D priors obtained from Tochilkin et al. (2024), COMPGS still enhances text-3D alignment, likely due to the dynamic optimization on composition-level Gaussians. Furthermore, compared with the optimization-based methods, COMPGS shows

Table 1: Quantitative comparisons on T³Bench He et al. (2023) and user studies show COMPGS outperforms feed-forward, optimization-based, and compositional generation models.

| Method | T³Bench (*Multiple Objects*) | | | User Study |
|---|---|---|---|---|
| | Quality↑ | Alignment↑ | Average↑ | Ranking Score↑ |
| OpenLRM Hong et al. (2023) | 15.2 | 25.5 | 20.4 | - |
| TripoSR Tochilkin et al. (2024) | 16.7 | 28.6 | 22.7 | - |
| DreamFusion Poole et al. (2022) | 17.3 | 14.8 | 16.1 | 4.54 |
| SJC Wang et al. (2023a) | 17.7 | 5.8 | 11.7 | 3.08 |
| LatentNeRF Metzer et al. (2023) | 21.7 | 19.5 | 20.6 | 4.14 |
| Fantasia3D Chen et al. (2023b) | 22.7 | 14.3 | 18.5 | 2.43 |
| ProlificDreamer Wang et al. (2024) | 45.7 | 25.8 | 35.8 | 3.56 |
| Magic3D Lin et al. (2023) | 26.6 | 24.8 | 25.7 | 4.30 |
| Set-the-Scene Cohen-Bar et al. (2023) | 20.8 | 29.9 | 25.4 | - |
| VP3D Chen et al. (2024c) | 49.1 | 31.5 | 40.3 | 6.71 |
| COMPGS (ours) | 54.2 (+5.1) | 37.9(+6.4) | 46.1(+5.8) | 7.23 |

significant advantages in both two metrics. Even when compared to methods specifically designed for compositional generation, e.g., VP3D Chen et al. (2024c) and Set-the-Scene Cohen-Bar et al. (2023), COMPGS demonstrates clear improvements, indicating the effectiveness of our designs. Note that COMPGS is not limited to compositional generation; the results in the track of single-object generation are included in Appendix A.4.

**Runtime Comparisons** Though COMPGS is trained with 10k steps, we observed that training the model for 5k iterations already produces high-quality content with minimal loss of texture details. We show runtime comparisons with other models in Tab. 2. In fact, compositional 3D generation (left) requires a longer training time than single-object generation (right) due to its complexity. Compositional 3D involves optimizing individual objects and ensuring the consistency of compositional scenes, making them more intricate. The runtime is generally relative to the number of objects involved. Compared to open-source compositional 3D methods such as Set-the-Scene Cohen-Bar et al. (2023), Progressive3D Cheng et al. (2023), and GraphDraemer Gao et al. (2024), our proposed COMPGS is more efficient in training. For instance, given the prompt *"a parrot talks beside a perch and two bowls,"* Progressive3D takes approximately 250 minutes for 3D generation, while Set-the-Scene requires around 110 minutes. Since many compositional scene generation methods are not open-sourced, we also present the training steps listed in the papers for straightforward comparisons. In addition, CompGS demonstrates comparable and even superior efficiency over other methods in text-to-single object generation. For example, methods such as Magic3D Lin et al. (2023) and Fantastic3D Chen et al. (2023b) require over five hours to optimize a single object. In contrast, CompGS can achieve the same task in approximately 30 minutes.

Table 2: Runtime comparisons on both compositional generation and single object generation show the efficiency of COMPGS.

| Method | Compositional Generation | | | | Method | Single Object Generation | | | |
|---|---|---|---|---|---|---|---|---|---|
| | Open-source | 3D Representations | Training Steps | Training Time (minutes) | | Open-source | 3D Representations | Training Steps | Training Time (minutes) |
| Progressive3D Cheng et al. (2023) | ✓ | NeRF | 40,000 | 220 | DreamFusion Poole et al. (2022) | ✓ | NeRF | - | 360 |
| Set-the-scene Cohen-Bar et al. (2023) | ✓ | NeRF | 15,000 | 110 | Magic3D Lin et al. (2023) | ✓ | NeRF | - | 340 |
| CompNeRF Driess et al. (2023) | ✗ | NeRF | 13,000 | - | Fantastic3D Chen et al. (2023b) | ✓ | NeRF | - | 380 |
| SceneWiz3D Zhang et al. (2024b) | ✗ | NeRF | 20,000 | 420 | ProlificDreamer Wang et al. (2024) | ✓ | NeRF | - | 520 |
| GraphDreamer Gao et al. (2024) | ✓ | NeRF | 20,000 | 420 | GaussianDreamer Yi et al. (2023) | ✓ | 3D Gaussians | - | 14 |
| CompGS-10k (Ours) | - | 3D Gaussians | 10,000 | 70 | CompGS-10k (Ours) | - | 3D Gaussians | 10,000 | 70 |
| CompGS-5k (Ours) | - | 3D Gaussians | 5,000 | 30 | CompGS-5k (Ours) | - | 3D Gaussians | 5,000 | 30 |

**Extended Applications: 3D Editing** COMPGS allows progressive 3D editing for compositional scenes. We present 3D editing examples in Fig. 5. These examples demonstrate that by transferring edited 2D compositionality and employing dynamic SDS optimization, COMPGS can progressively incorporate new 3D entities into the original 3D scenes. For example, COMPGS can generate the *'chair'*, *'panda'*, *'hat'* and *'plant'* step by step as shown in Fig. 5.

Figure 5: 3D Editing examples of COMPGS. More examples could be found in Appendix A.9.

Figure 6: Visual results of the ablation studies on three key designs in COMPGS.

## 4.3 ABLATION STUDY

We conduct ablation studies to validate the effectiveness of three key designs in COMPGS: Gaussian parameters initialization, decomposed optimization (DO), and volume-adaptive optimization (VAO). We randomly choose 20 prompts from $T^3$Bench for the quantitative evaluation. Meanwhile, we use Fig. 6 to visualize the effect of each component.

**Initialization** Instead of initializing Gaussian parameters using 2D compositionality, we conduct random initialization within a predefined 3D bounding box for each entity Gaussian. A significant decrease in the quality and alignment metric is observed in Tab. 3. Besides, the visualization in Fig. 6 confirms that training from random initialization with predefined layouts may result in low-quality textures (e.g., owl's face) and missing entities (e.g., the branch).

Table 3: Ablation Studies on $T^3$Bench He et al. (2023).

| Component | Quality | Alignment | Average |
|---|---|---|---|
| Full Setting | 53.8 | 38.0 | 45.9 |
| - w/o GS Init. | 22.8 | 18.7 | 20.8 |
| - w/o DO Strategy | 46.8 | 35.2 | 41.0 |
| - w/o VAO Strategy | 50.8 | 36.4 | 43.6 |

**Decomposed Optimization** We analyze the decomposed training by discarding the entity-level optimization. Results in Tab. 3 and Fig 6 both indicate that removing decomposed training leads to low-fidelity generation. For example, in Fig. 6, optimizing composition-level Gaussians results in a relatively blurred generation on the owl's fur and the pinecone, validating that optimization on each single entity is necessary to effectively leverage 2D diffusion priors to guide 3D generation.

**Volume-adaptive Optimization** To evaluate the effectiveness of volume-adaptive optimization, we choose to optimize the entity Gaussians in their original 3D space. We observed that optimizing different entities within varying sizes of 3D space slightly decreased in quality and alignment, which is noticeable for smaller objects in the compositional scene. For instance, the *'pinecone'* and *'branch'* trained without volume-adaptive optimization exhibited fewer fine-grained details, shown as the green leaves on the branch and the shadow on the pinecone in Fig. 6.

## 5 CONCLUSION

In this paper, we introduced COMPGS, a user-friendly, optimization-based framework, which achieves compositional text-to-3D generation utilizing Gaussian Splatting. Our three core designs, including initializing 3D Gaussians with 2D compositionality, decomposed optimization for either entity-level or composition-level Gaussians, and volume-adaptive optimization, contribute to the success of COMPGS. Through both qualitative and quantitative experiments, we demonstrate COMPGS is capable of generating complex interactions between multiple entities in 3D scene. We hope that our work inspires further innovation and advancements in the compositional 3D generation.

**Impacts and Limitations** As a text-to-3D model, we do not foresee obvious undesirable ethical/social impacts. One limitation is that when the given prompt includes backgrounds (e.g., ground, sky), COMPGS may fail to generate these elements adequately. This is due to the current text-guided segmentation model's inability to segment such abstract concepts. We leave it for future exploration.

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

# A    APPENDIX / SUPPLEMENTAL MATERIAL

In this supplementary material, we first clarify the notations used in this paper and then revisit the proposed COMPGS in Algorithms 1. The training details of COMPGS will also be provided. Besides, we provide more numerical and visual evaluations to further validate the effectiveness of our model. We have provided **a demo video** in the attachment to display more visual comparisons between COMPGS and other methods. We will make code public.

## A.1    NOTATIONS

We compile a comprehensive list of all the notations utilized in this paper, as shown in Table 4.

Table 4: Notations.

| Notation | Description |
|----------|-------------|
| $L$ | Total number of entities |
| $V$ | Complex prompt (e.g., 'an owl perches on a branch near a pinecone') |
| $I$ | Composed image generated by the 2D diffusion model |
| $v_l$ | Entity-level prompt for entity $l, (l \in L)$ |
| $I_l$ | Segmented image containing entity $l, (l \in L)$ |
| $m_l$ | Rough triangle mesh of the 3D entity $l, (l \in L)$ |
| $\theta_l$ | 3D Gaussians for the entity $l, (l \in L)$ |
| $\theta$ | Composed 3D Gaussians $l$ |
| $N$ | Number of points indexed from each mesh |
| $\mu_i^l$ | Center positions of each vertex of mesh $m_l$ in $\mathbb{R}^3$ |
| $c_i^l$ | Texture colors queried from each vertex of mesh $m_l$ in $\mathbb{R}^3$ |
| $\text{bbox}_l$ | 3D bounding box for entity $l$, used for optimization |
| $\text{bbox}_{std}$ | Standardized volumetric space for scaling |
| $\mu$ | Center positions of each vertex in the original 3D space |
| $\hat{\mu}$ | Transformed center positions of entity Gaussian after scaling |
| $\beta$ | Shift parameters for the center positions of the bounding box |
| $\lambda$ | Scale parameters for standardizing the volumetric space |
| x | Rendered image from 3D Gaussians |
| $g(\cdot)$ | Gaussian Splatting rendering function |
| $\beta$ | the shift parameters for volume-adaptive optimization |
| $\lambda$ | the scale parameters for volume-adaptive optimization |
| $\text{Mean}(\cdot)$ | the operator computing the center coordinates of the given bounding box |
| $\hat{\theta}$ | New Gaussians initialized from the edited 2D image |

## A.2    ALGORITHM

We provide pseudocode in Algorithm 1. Two core designs, including 3D Gaussian initialization with 2D compositionality and dynamic SDS optimization, are detailed.

## A.3    ADDITIONAL TRAINING DETAILS

COMPGS is implemented in ThreeStudio Guo et al. (2023). We use DALL·E 3 Betker et al. (2023), LangSAM Medeiros (2024) and TripoSR Tochilkin et al. (2024) to implement the text-to-image, text-guided segmentation, and image-to-mesh, respectively. For entity-level optimization, we adopt MVDream Shi et al. (2023) as the 3D diffusion prior; while for composition-level optimization, we employ *stabilityai/stablediffusion-2-1-base* Rombach et al. (2022b) as the 2D diffusion prior. We set all the diffusion guidance as 50. For all Gaussian parameters, we linearly decreased the learning rate for position $\mu$ from $10^{-3}$ to $10^{-5}$, for scale from $10^{-2}$ to $10^{-3}$, and for color $c$ from $10^{-2}$ to $10^{-3}$, respectively. Besides, we fixed the learning rate for opacity $a$ to be 0.05, and for rotation to be 0.001. Additionally, we use a consistent batch size of 4 for both training and test, and a rendered resolution fixed at $1024 \times 1024$. Camera settings during training are set with distances ranging from 0.8 to 1.0 relative units, a field of view between 15 and 60 degrees, and elevation ranging up to 30 degrees. Additionally, there are no perturbations applied to camera position, center, or orientation, maintaining a controlled imaging environment. For test, we set the resolution of

**Algorithm 1** COMPGS: 3D Gaussian Initialization and Dynamic SDS Optimization $V, \{v_l\}(l \in L)$: Input prompt and entity-level prompts.
$\{m_l\}(l \in L)$: Entity-level meshes.
$\theta, \{\theta_l\}(l \in L)$: Composition-level Gaussian parameters and entity-level Gaussian parameters.
$\text{bbox}_{std}$: Standardized volumetric space.
$L$: The number of entities.
$N$: The number of Gaussian parameters.
T2I: Text-to-Image models.
TGS: Text-guided segmentation models.
I2M: Image-to-Mesh models.
$\text{Zoom}^{\uparrow}, \text{Zoom}^{\downarrow}$: Zoom-in and Zoom-back operators in Eq. 4.
$\eta$: Learning rate.
$T$: Total training iterations.

---

*Stage 1: Initializing 3D Gaussians with 2D Compositionality.*
$I = \text{T2I}(V)$                          ▷ Generate well-composed Image from the given prompt
$\{v_l\} = \text{LLM}(V)$                           ▷ Obtain entity-level prompts via LLM
$\{m_l\} = \text{I2M}(\text{TGS}(\{v_l\}, I))$                    ▷ Obtain entity-level meshes
$\mu_i(i \in N), c_i(i \in N) \leftarrow m_l(l \in L)$            ▷ Positions and colors of the 3D Gaussians.
$D \leftarrow \mu_i(i \in N)$                     ▷ Distance between the nearest two positions.
$\Sigma_i(i \in N), \alpha_i(i \in N) \leftarrow D, 0.1$             ▷ Covariance and opacity of the 3D Gaussians.
$\text{bbox}_l(l \in L) \leftarrow \mu_i(i \in N)$                   ▷ Boundary of bounding box

*Stage 2: Dynamic SDS Optimization.*
**for** $t = 1$ to $T$ **do**
    $l \leftarrow \text{randint}(1, L)$              ▷ Randomly select an integer $l$ from the range 1 to $L$
    **if** $i = 0$ **then**
$$\nabla_\theta \mathcal{L}_{\text{SDS}}^{2d}(\phi, \text{x} = g(\theta)) \triangleq \mathbb{E}_{t,\epsilon}\left[w(t)\left(\hat{\epsilon}_\phi(\mathbf{z}_t, V, t) - \epsilon\right)\frac{\partial \text{x}}{\partial \theta}\right]$$
                         ▷ Obtain the gradients via SDS loss with 2D priors
$$\nabla_\theta \mathcal{L}_{\text{SDS}}^{3d}(\phi, \text{x} = g(\theta)) \triangleq \mathbb{E}_{t,\epsilon}\left[w(t)\left(\hat{\epsilon}_\phi(\mathbf{z}_t, v, t) - \epsilon\right)\frac{\partial \text{x}}{\partial \theta}\right]$$
                         ▷ Obtain the gradients via SDS loss with 3D priors
        $\theta \leftarrow \theta - \eta(\nabla_\theta \mathcal{L}_{\text{SDS}}^{2d} + \nabla_\theta \mathcal{L}_{\text{SDS}}^{3d})$
                 ▷ Update the compositional Gaussian parameters via back-propagation
    **else**
        $\hat{\theta}_l \leftarrow \text{Zoom}^{\uparrow}(\theta_l, \text{bbox}_l, \text{bbox}_{std})$
    ▷ Dynamically zoom-in Gaussian parameters from $\text{bbox}_l$ to a standardized space $\text{bbox}_{std}$
$$\nabla_{\hat{\theta}_l} \mathcal{L}_{\text{SDS}}^{3d}(\phi, \text{x} = g(\hat{\theta}_l)) \triangleq \mathbb{E}_{t,\epsilon}\left[w(t)\left(\hat{\epsilon}_\phi(\mathbf{z}_t, v_l, t) - \epsilon\right)\frac{\partial \text{x}}{\partial \hat{\theta}_l}\right]$$
                         ▷ Obtain the gradients via SDS loss with 3D priors
        $\hat{\theta}_l \leftarrow \hat{\theta}_l - \eta \nabla_{\hat{\theta}_l} \mathcal{L}_{\text{SDS}}^{3d}$
                 ▷ Update the compositional Gaussian parameters via back-propagation
        $\theta_l \leftarrow \text{Zoom}^{\downarrow}(\hat{\theta}_l, \text{bbox}_l, \text{bbox}_{std})$
    ▷ Dynamically zoom-back Gaussian parameters from the standardized space $\text{bbox}_{std}$ to $\text{bbox}_l$

    **end for**

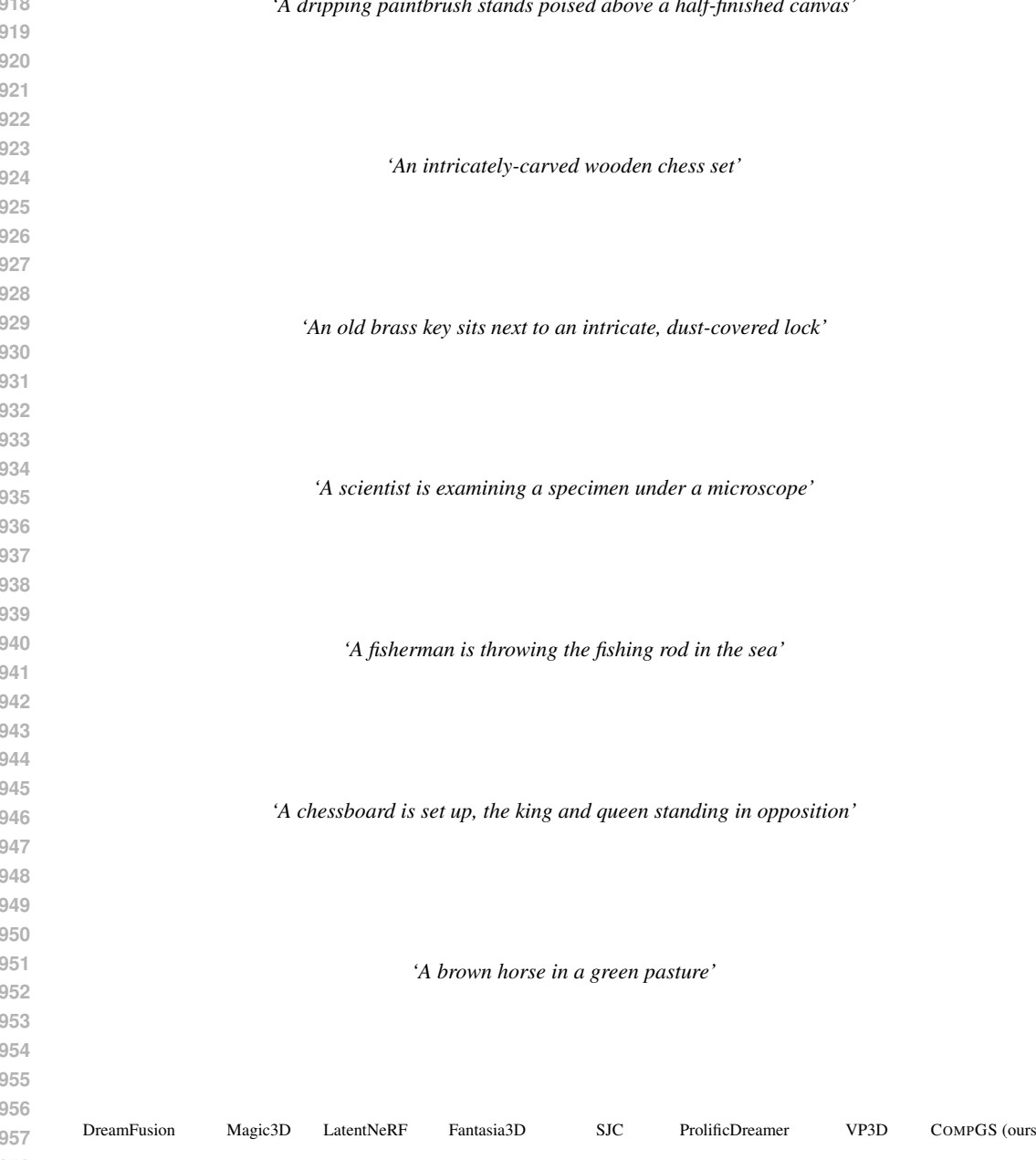

*'A dripping paintbrush stands poised above a half-finished canvas'*

*'An intricately-carved wooden chess set'*

*'An old brass key sits next to an intricate, dust-covered lock'*

*'A scientist is examining a specimen under a microscope'*

*'A fisherman is throwing the fishing rod in the sea'*

*'A chessboard is set up, the king and queen standing in opposition'*

*'A brown horse in a green pasture'*

DreamFusion     Magic3D     LatentNeRF     Fantasia3D     SJC     ProlificDreamer     VP3D     COMPGS (ours)

Figure 7: **Qualitative comparisons between COMPGS and other text-to-3D models on** $\text{T}^3$**Bench (multiple objects track).** COMPGS is better at generating highly-composed, high-quality 3D contents that strictly align with the given texts. Watch the animations by **clicking** them (Not all PDF readers support playing animations. Best viewed in Acrobat/Foxit Reader).

rendered image as $1024 \times 1024$ with specific camera distance and field of view for validation set to 3.5 units and 40 degrees, respectively. For each prompt, we train the model on an NVIDIA A100 GPU (40G) for 10,000 iterations, which takes approximately 70 minutes. We observed that training the model for 5,000 iterations already produces high-quality content with minimal loss of texture details. This indicates that the training duration can be shortened to around **30 minutes**. However, to achieve high-quality 3D textures, we use 10,000 iterations for training in this paper, unless otherwise specified.

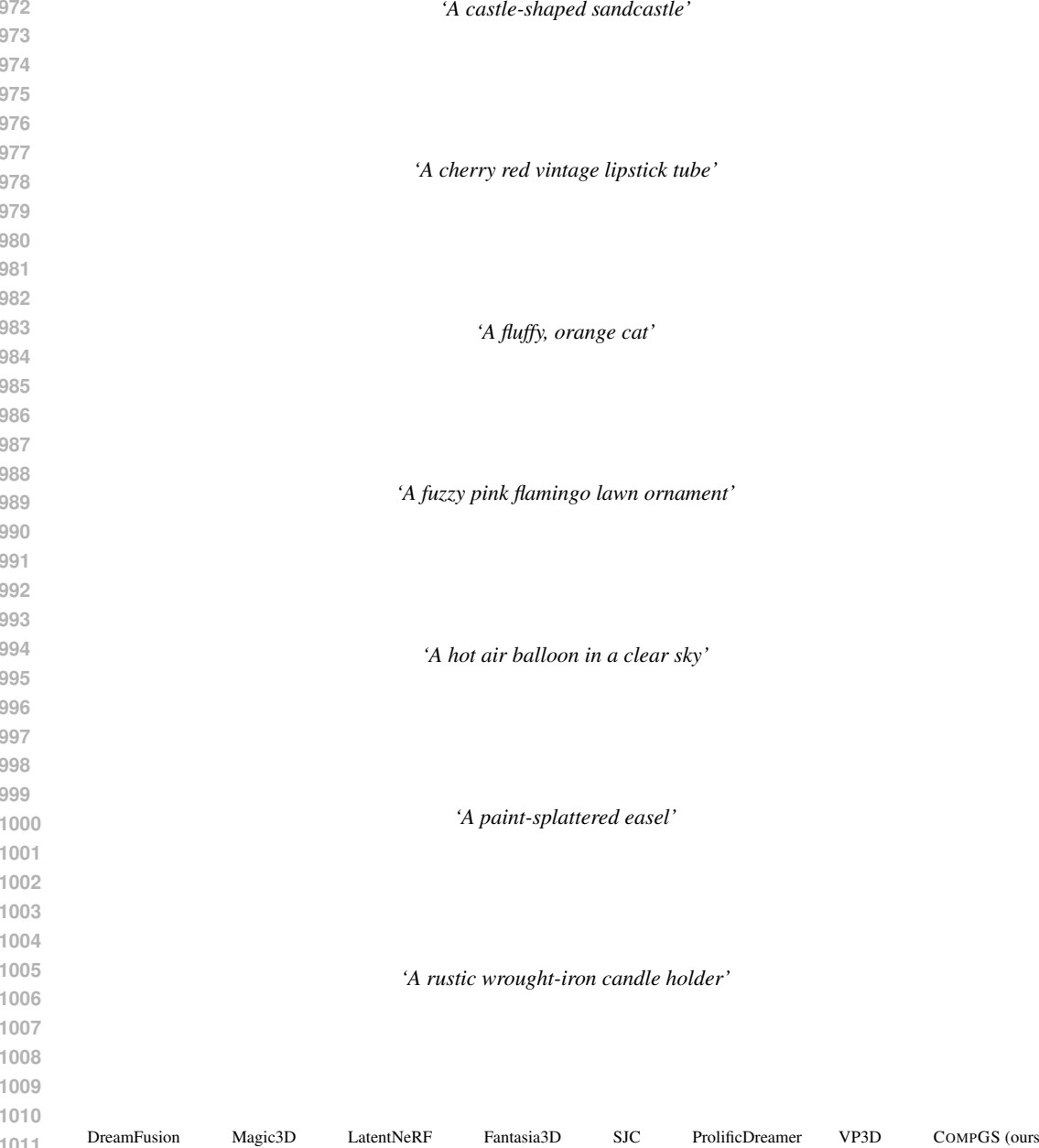

*'A castle-shaped sandcastle'*

*'A cherry red vintage lipstick tube'*

*'A fluffy, orange cat'*

*'A fuzzy pink flamingo lawn ornament'*

*'A hot air balloon in a clear sky'*

*'A paint-splattered easel'*

*'A rustic wrought-iron candle holder'*

DreamFusion    Magic3D    LatentNeRF    Fantasia3D    SJC    ProlificDreamer    VP3D    COMPGS (ours)

Figure 8: **Qualitative comparisons between COMPGS and other text-to-3D models on** $\mathrm{T}^3$**Bench (single object track).** COMPGS is better at generating high-quality 3D assets that strictly align with the given texts. Watch the animations by **clicking** them (Not all PDF readers support playing animations. Best viewed in Acrobat/Foxit Reader).

## A.4    EXTENDED EXPERIMENTS ON QUALITATIVE COMPARISONS

**Qualitative Model Comparisons on Multi-objects Generation** Fig. 7 showcases additional 3D assets produced by COMPGS. The prompts are selected from $\mathrm{T}^3$Bench (multiple objects track). Compared to previous methods, COMPGS not only generates multiple objects but also produces more plausible interactions while maintaining 3D consistency among the objects. For example, in the first row, previous methods such as DreamFusion, Magic3D, LatentNeRF, Fantasia3D, SJC, and ProlificDreamer all fail to generate the canvas described in the given prompt. Although both VP3D and COMPGS can generate the two entities (paintbrush and canvas), VP3D fails to maintain 3D consistency, as the back view of the canvas is not visually plausible. In this case, COMPGS

Table 5: **Quantitative comparisons with baselines on T³Bench He et al. (2023) (all three tracks).** COMPGS is compared with feed-forward models, optimization-based models, and models specifically designed for compositional generation.

| Method | Single Object | | | Single Object with Surroundings | | | Multiple Objects | | |
|---|---|---|---|---|---|---|---|---|---|
| | Quality | Alignment | Average | Quality | Alignment | Average | Quality | Alignment | Average |
| LRM Hong et al. (2023) | 29.4 | 38.2 | 33.8 | 20.3 | 35.1 | 27.7 | 15.2 | 25.5 | 20.4 |
| TripoSR Tochilkin et al. (2024) | 34.3 | 38.9 | 36.6 | 21.8 | 37.2 | 29.5 | 16.7 | 28.6 | 22.7 |
| DreamFusion Poole et al. (2022) | 24.9 | 24.0 | 24.4 | 19.3 | 29.8 | 24.6 | 17.3 | 14.8 | 16.1 |
| SJC Wang et al. (2023a) | 26.3 | 23.0 | 24.7 | 17.3 | 22.3 | 19.8 | 17.7 | 5.8 | 11.7 |
| LatentNeRF Metzer et al. (2023) | 34.2 | 32.0 | 33.1 | 23.7 | 37.5 | 30.6 | 21.7 | 19.5 | 20.6 |
| Fantasia3D Chen et al. (2023b) | 29.2 | 23.5 | 26.4 | 21.9 | 32.0 | 27.0 | 22.7 | 14.3 | 18.5 |
| ProlificDreamer Wang et al. (2024) | 51.1 | 47.8 | 49.4 | 42.5 | 47.0 | 44.8 | 45.7 | 25.8 | 35.8 |
| Magic3D Lin et al. (2023) | 38.7 | 35.3 | 37.0 | 29.8 | 41.0 | 35.4 | 26.6 | 24.8 | 25.7 |
| Set-the-Scene Cohen-Bar et al. (2023) | 32.9 | 31.9 | 32.4 | 30.2 | 45.8 | 35.5 | 20.8 | 29.9 | 25.4 |
| VP3D Chen et al. (2024c) | 54.8 | 52.2 | 53.5 | 45.4 | 50.8 | 48.1 | 49.1 | 31.5 | 40.3 |
| COMPGS | 55.1 | 52.5 | 53.8 | 43.2 | 46.8 | 45.0 | 54.2 | 37.9 | 46.1 |

successfully captures both the key entities described in the prompt and generates reasonable spatial relationships and interactions between the two objects. This phenomenon can also be observed in other cases, such as the key and lock in the third row, and the fisherman in the sea in the fifth row, and so on. Besides the issue of 3D consistency, we found that COMPGS performs better in texture alignment. For example, in the second-to-last row, other methods failed to display the combination of chessboard, king, and queen. Specifically, VP3D did not recognize the king and queen as chess pieces. In contrast, COMPGS generates these entity details more accurately. Overall, the comparisons in both visual quality and textural alignment with previous methods demonstrate the effectiveness of the proposed COMPGS.

**Qualitative Model Comparisons on Single-object Generation** Though COMPGS is specifically designed for compositional generation, it can naturally handle single-object generation as well. We present the qualitative comparisons between COMPGS and previous works in Fig. 8. It is observed that COMPGS performs better in maintaining multi-view consistency and generating fine-grained details of the object. For example, in the last row of Fig. 8, COMPGS is capable of generating a 3D consistent candle holder, including detailed copper textures. In contrast, other methods either fail to produce the corresponding shape Chen et al. (2023b), only generate rough outlines without detailed textures Poole et al. (2022); Lin et al. (2023); Metzer et al. (2023); Wang et al. (2023a), or produce 3D patterns with discontinuities Wang et al. (2024); Chen et al. (2024c).

**Qualitative Model Comparisons with Scene-generation Methods** We also compare COMPGS with closed-source models Zhou et al. (2024); Cohen-Bar et al. (2023) that generate 3D scenes. Figures were selected from Zhou et al. (2024) and are presented in Fig. 9. The results indicate that COMPGS excels in generating high-fidelity texture details and complex interactions. In the second row of Fig. 9, COMPGS produces more detailed textures for table legs and rabbit fur. Regarding interaction generation, Set-the-Scene Cohen-Bar et al. (2023) fails to create complex spatial relationships, as shown with the dog and the Great Pyramid in the first row. Although GALA3D can generate reasonable spatial relationships, it fails to incorporate mutual interactions between objects. This is because it performs compositional generation by optimizing the layout of each object individually, neglecting other inter-interactions such as the rabbit's mouth on the cake and the dog's paw on the plate. In contrast, COMPGS generates higher-fidelity textures (e.g., the table body, rabbit fur) and more realistic interactions among objects (e.g., the dog's paw hanging off the plate rather than just resting on top).

**Qualitative Model Comparisons with Other Compositional Generation Methods** In the main paper, we have compared COMPGS with both open-sourced compositional 3D generation baselines (Set-the-scene and VP3D) in Table 1, and close-sourced baselines (GALA3D) in Figure 9. Results show that the 3D assets generated by COMPGS are not only high-quality in appearance, but also align with the given prompts more strictly. We have included qualitative comparisons with both GraphDreamer Gao et al. (2024) and DreamGaussian Tang et al. (2023) in Fig. 10. Results show that COMPGS demonstrates superior performance on both generation quality and text-3d alignment.

### A.5 QUANTITATIVE MODEL COMPARISONS

Tab. 5 presents the complete quantitative comparisons on all three tracks of T³Bench. The results indicate that COMPGS achieved state-of-the-art performance in compositional generation and slightly outperformed competitors in the single object track. For instance, in the multiple object track, our model surpassed the second-best work Chen et al. (2024c) by 5.1 in quality and 6.4 in texture

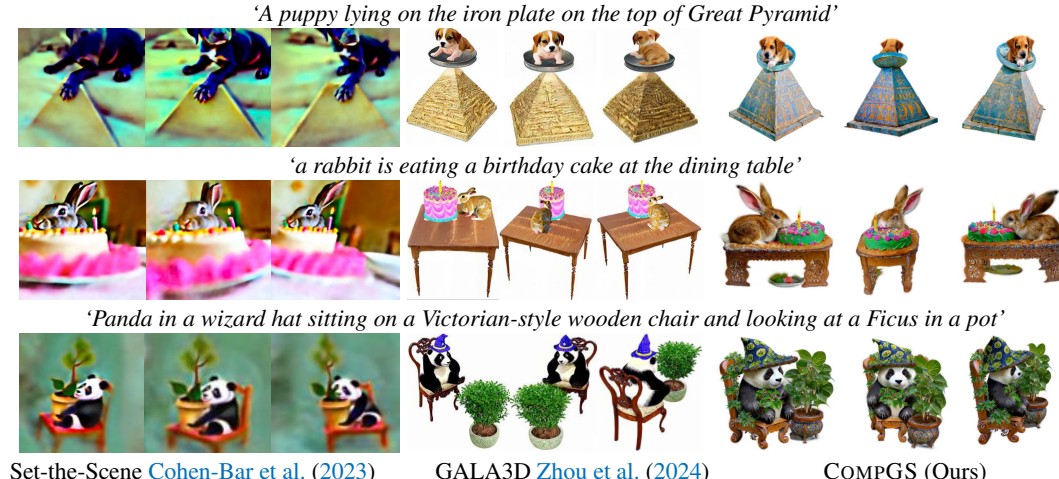

'A puppy lying on the iron plate on the top of Great Pyramid'

'a rabbit is eating a birthday cake at the dining table'

'Panda in a wizard hat sitting on a Victorian-style wooden chair and looking at a Ficus in a pot'

Set-the-Scene Cohen-Bar et al. (2023)  GALA3D Zhou et al. (2024)  COMPGS (Ours)

Figure 9: **Qualitative Comparisons Between COMPGS and 3D Scene Generation Methods.** We selected the figures from Zhou et al. (2024) for these comparisons due to the unavailability of the code. COMPGS performs better in generating object textures and complex interactions.

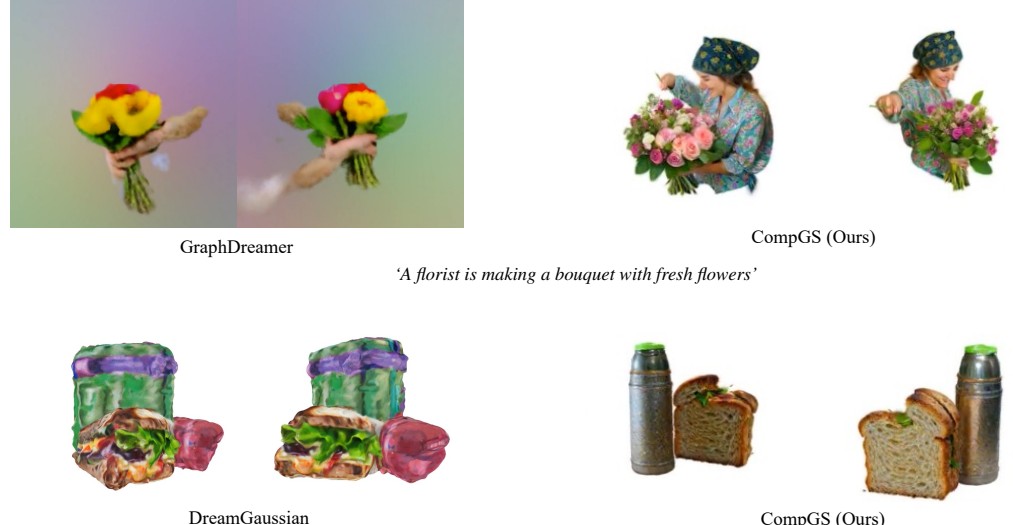

GraphDreamer                                        CompGS (Ours)

'A florist is making a bouquet with fresh flowers'

DreamGaussian                                      CompGS (Ours)

'A half-eaten sandwich sits next to a lukewarm thermos'

Figure 10: Extended comparisons with GraphDreamer Gao et al. (2024) and DreamGaussian Tang et al. (2023).

alignment. In the single object track, our model also slightly outperformed the second-best work Chen et al. (2024c) by 0.3 in both quality and alignment.

However, it is worth noting that our model did not achieve state-of-the-art performance in generating single objects with surroundings. This is attributed to the text-guided segmentation model we use, which does not effectively segment the background (e.g., ground, sky, etc.). We have explained this in Sec. 5 and leave it for future improvement. Despite a slight decline in our texture alignment metric in this track, our model still performed significantly better than other methods Tochilkin et al. (2024); Hong et al. (2023); Poole et al. (2022); Wang et al. (2023a); Metzer et al. (2023); Chen et al. (2023b); Wang et al. (2024); Lin et al. (2023); Cohen-Bar et al. (2023), except for VP3D.

### A.6 Examples in User Study

We provide examples of images and scenes used in our user study. In particular, we present concatenated rendering videos and ask participants to rank the eight methods shown in the video based on the overall quality of the 3D objects and the alignment between the text and the 3D models. We average the rank number as its ranking score for comparisons in Tab. 1.

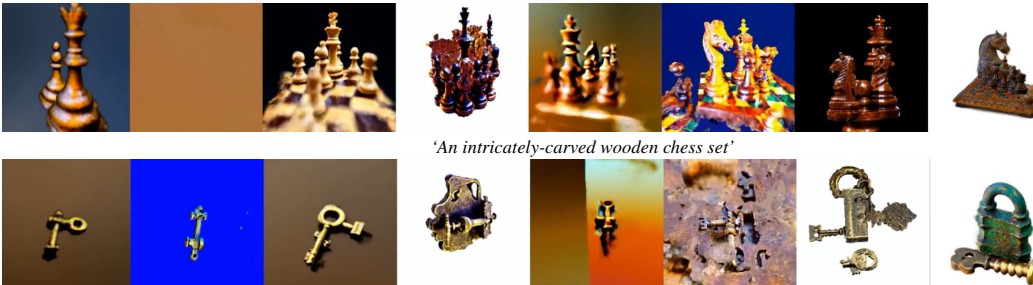

*'An intricately-carved wooden chess set'*

*'An old brass key sits next to an intricate, dust-covered lock'*

Figure 11: Examples used in our user study.

### A.7 Robustness

We empirically found that COMPGS demonstrates the ability to address certain deficits caused by off-the-shelf model priors (e.g., T2I and segmentation priors). Here are some illustrative examples: (1) If certain parts of the target objects are not correctly segmented, COMPGS can complete the unsegmented part with correct 3D information. This is demonstrated in Fig. ???12(left), where the swing has not been segmented but has been generated by COMPGS correctly. This is facilitated through the Entity-level Optimization procedure proposed in the DO strategy. (2) If the T2I models fail to generate proper intra-object interactions, COMPGS can correct the multi-object interactions. This is shown in Fig. ???12(right), where the spatial relationships in the given image are incorrect and then corrected in the text-to-3D process. This is achieved by the Composition-level Optimization in the proposed DO strategy.

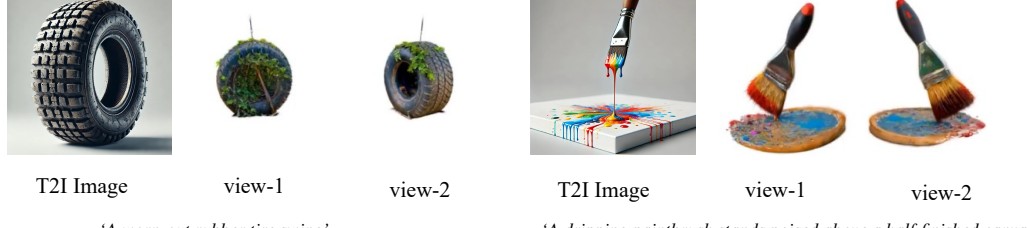

T2I Image     view-1     view-2     T2I Image     view-1     view-2

*'A worn-out rubber tire swing'*     *'A dripping paintbrush stands poised above a half-finished canvas'*

Figure 12: CompGS demonstrates the ability to address certain deficits caused by off-the-shelf priors.

### A.8 Failure Cases

As discussed in Sec. 5, COMPGS exhibits limitations in generating backgrounds, such as ground and sky. This is likely due to the current text-guided segmentation model's inability to effectively segment these abstract concepts. When the background is not well-segmented, we lose the corresponding 2D compositionality needed for initializing 3D Gaussians. This leads to two failure cases: (1) the absence of background in the compositional 3D scenes, as seen with the missing grass in the second column of Fig. 13, or (2) background generation of poor visual quality, such as the vague and unclear depiction of grass in the first column of Fig. 13. It's crucial to note that such limitations, whilst exist, are not the focus of this work. These shortcomings can be overcome by enhancing the capabilities of off-the-shelf models, effectively mitigating the manifested issues.

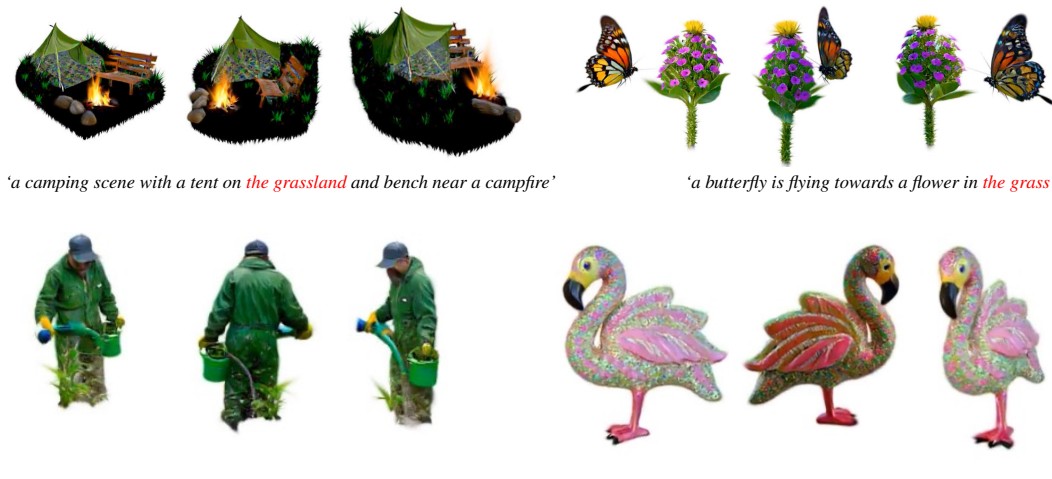

*'a camping scene with a tent on the grassland and bench near a campfire'*          *'a butterfly is flying towards a flower in the grass'*

*'A gardener is watering plants with a hose'*          *'A fuzzy pink flamingo lawn ornament on the water'*

Figure 13: **Failure Cases of COMPGS in background generation** When text-guided segmentation mode fails to segment the backgrounds, COMPGS may generate background with poor visual quality or fails to generate background.

### A.9    3D EDITING EXAMPLES

COMPGS offers a user-friendly approach to progressively conduct 3D editing for compositional 3D generation. More visual examples are presented in Fig. ???14. For instance, given a compositional prompt such as *'A puppy lying on the iron plate on the top of the Great Pyramid, with a pharaoh nearby'*, we divide the generation process into four stages. Initially, we generate *'the Great Pyramid'* on the left, then progressively add *'the plate'*, *'the puppy'*, and *'the pharaoh'* to complete the 3D scene. Notably, both the interactions and texture details can be well-produced during the editing pipeline of COMPGS.

Overall prompt: *'an owl perches on a branch near a pinecone, with a rat below the branch'*

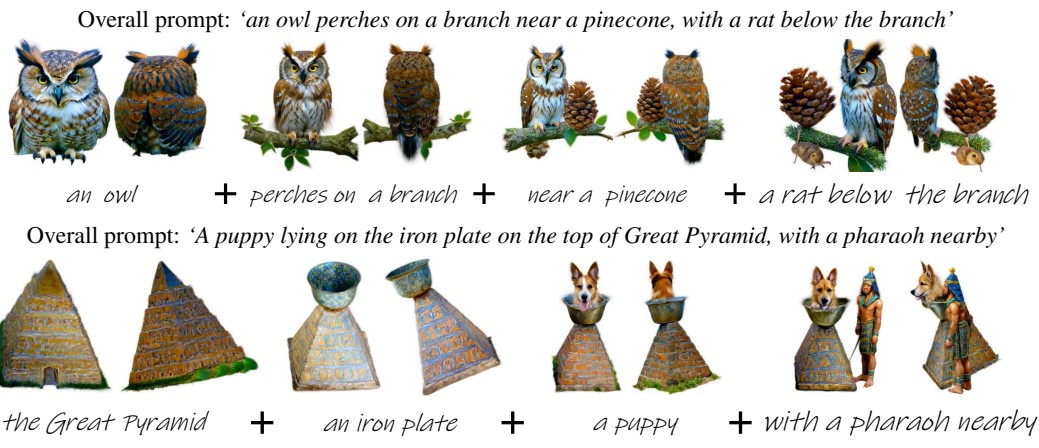

Figure 14: **More examples of 3D Editing.** COMPGS provides a user-friendly way to progressively edit on 3D scenes for compositional generation.

