# OpenReview forum: "CompGS: Unleashing 2D Compositionality for Compositional Text-to-3D via Dynamically Optimizing 3D Gaussians"
_ICLR.cc/2025/Conference — ICLR 2025 Conference Withdrawn Submission_

### Official Review · Reviewer_dRVf · 2024-10-29

**Soundness:** 3
**Presentation:** 4
**Contribution:** 2
**Rating:** 5
**Confidence:** 3

**Summary:**

This paper introduces CompGS, a method for generating 3D scenes involving multiple objects with complex interactions. The core design includes the initialization of 3D Gaussians with transferred 2D compositionality, alternating between entity-level and composition-level optimization, and dynamically maintaining volume consistency during entity-level optimization.

**Strengths:**

1. CompGS transfers 2D compositionality to facilitate composed 3D generation. Unlike previous methods such as GALA3D, which directly use LLM to provide layouts, this approach is more beneficial for initializing the Gaussians of objects with complex mutual interactions.
2. The proposed method is well-explained, and the experimental details are clearly presented. The approach is well-motivated, addressing real issues such as generating complex mutual interactions among objects and optimizing small objects during compositional 3D generation.
3. The experimental results are impressive, particularly the generated 3D humans and animals, which do not seem to suffer from the multi-view Janus Problem.

**Weaknesses:**

1. The paper may not introduce many novel insights compared to previous methods. The dynamic optimization strategy of CompGS and the compositional optimization strategy in GALA3D[1] appear to be similar. Both alternate between composition-level optimization and entity-level optimization. GALA3D optimizes a single entity with a dynamic camera radius of $\frac{3}{4}\left\|\left(h_i, w_i, l_i\right)\right\|_2$ , which has an effect similar to the Volume-adaptively Optimizing Entity-level Gaussians in CompGS. Furthermore, the initialization of 3D Gaussians with 2D compositionality, compared to methods like REPARO[2] and ComboVerse[3], seems to add only an additional step. This step involves using 2D diffusion models to generate a well-composed image from a given prompt and obtaining entity-level prompts via LLM.

2. Semantic drift and pattern leakage: In Figure 5 of the appendix, the chair and hat exhibit leaf-like patterns similar to those of the ficus in a pot.

References:

[1] Zhou, Xiaoyu, et al. "Gala3d: Towards text-to-3d complex scene generation via layout-guided generative gaussian splatting." arXiv preprint arXiv:2402.07207 (2024).

[2] Han, Haonan, et al. "REPARO: Compositional 3D Assets Generation with Differentiable 3D Layout Alignment." arXiv preprint arXiv:2405.18525 (2024).

[3] Chen, Yongwei, et al. "Comboverse: Compositional 3d assets creation using spatially-aware diffusion guidance." arXiv preprint arXiv:2403.12409 (2024).

**Questions:**

1. In the initialization of 3D Gaussians with 2D compositionality, how is the depth of entities determined, i.e., how is the distance from the entities to the screen calculated?
2. For entity-level optimization, in Equation (4), it seems that only translation and scaling are performed. Is rotation applied? How does CompGS handle entity-level optimization when the front view of an entity is not facing the front of the scene, or when the front views of two objects in the same scene differ by more than 90 degrees?

---

### Official Review · Reviewer_YM4z · 2024-10-31

**Soundness:** 3
**Presentation:** 3
**Contribution:** 2
**Rating:** 5
**Confidence:** 4

**Summary:**

This paper introduces CompGS, a method for compositional text-to-3D scene generation. It features 3D Gaussian Initialization combined with 2D compositionality and Dynamic Optimization to improve performance. Experimental results demonstrate that this approach outperforms the baselines.

**Strengths:**

(1) The paper is well-written and easy to follow.
(2) The proposed method is simple and easy to understand. Sufficient details are provided for the readers.

**Weaknesses:**

(1) My concern about this paper is its limited novelty. While the results appear promising, I feel this paper may not provide substantial new insights for the readers.

(2) There is a lack of discussion regarding object interactions. In scenes with multiple objects, interactions are inevitable; however, the paper does not address how to manage complex interactions. For instance, if one object is obstructed by another in the image generated by T2I method, how should we effectively initialize and optimize the obstructed object?

(3) Some blurriness is noticeable in the results (for example, "An artistis painting on a blank canvas" in Fig 3, "a student is typing on his laptop" in Fig 4). The image quality falls short compared to other image-to-3D works, such as DreamCraft3D and DreamCraft3D++.

**Questions:**

After getting the image by T2I model, how do you initialize the center position for each object? Also, do you optimize object's position, scale, and rotation during optimization?

---

### Official Review · Reviewer_9hU3 · 2024-10-31

**Soundness:** 4
**Presentation:** 3
**Contribution:** 2
**Rating:** 5
**Confidence:** 5

**Summary:**

This paper targets at compositional 3D object generation. The paper proposes an optimization-based pipeline to take advantage on off-the-shelf models like 2D generation model, image-to-3D model, LLM, and explicit 3DGS representation.

**Strengths:**

Overall, the paper is well-written and easy to follow. Most of the components are clearly described and explained. The visual results shown are of high quality. Reasonable quantitative comparisons and ablations are provided.

**Weaknesses:**

# Unclear components
- LLM: Using LLM to provide priors for compositional generation is currently a common approach (the most similar use case to this paper is SceneWiz3D ). However, some details are still desirable. For examples, what is the prompt? What is the usual success rate? Any failure case?
- Editing: Is every component trained or only the added part? If they are all trained, how to keep the existing content unchanged? If only the added part trained, how to make sure it is compatible to existing components (as some editing might change existing part)?


# Comparison to Gala3D
- The paper mentioned the code/weight are not available. However, Gala3D does provide the code/weight via application.
- In L1061, the paper states "(Gala3D) fails to incorporate mutual interaction". I have reservations about the statement. For example, their teaser sample where an astronaut sitting on a couch, the interaction between the people and the couch is not trivial.
-  One straightforward adaptation and ablation is to replace the LLM-provided 3D bounding boxes with the 2D+img-to-3D bounding boxes, w/ and w/o using the 3D as initialization.

# Limitation/Issues of the pipeline
There are some evident limitations not discussed.
- Semantic leak due to 2D model
1. Fig 9, last row: the leaf leaks to the hat, panda, and the couch
- 3D inconsistency and blurry boundary, due to image-to-3D and optimization
1. Fig 9, the rabbit: The blurring stuff under the table. And it looks very 3d inconsistent.
2. Fig 4, top-left: the left-most viewpoint, the boundary between the cake and the gift is blurred. Also, it looks very 3d inconsistent.
- Occlusion due to  2d/3d ambiguity
1. Fig3, last row: what does two silken look like separately?
- Number of objects: The samples provided have number of objects less the five. In comparisons, Gala3D showed quite some samples with a larger amount of objects. Are there any hard constraint preventing the author to try the same prompts?

# Contribution:
- Explicit representation: Replacing the representation itself it not a contribution.
1. For example, how would set-the-scene perform if we replace its representation to GS?
2. L162 mentions SceneWiz3D is based on NeRF. However, they use NeRF for background and use meshes for foreground objects.
- local-global optimization: It is not new in composition-aware generation, for example, set-the-scene adopt similar strategy. Some discussions and comparison are desirable.

# MISC:
- Runtime in table 2: Are these results calibrates by number of object? Moreover, some of these methods include background, which is definitely an important factor to distinguish object-level and scene-level. This factor should also be considered in the table.

**Questions:**

All questions are listed above in the weakness section.
Overall:
- unclear component description regarding the LLM part and editing part.
- Insufficient comparisons to Gala3D.
- Limitations not discussed.
- Some parts of the contributions not discussed enough.

---

### Official Review · Reviewer_oYz8 · 2024-11-03

**Soundness:** 3
**Presentation:** 3
**Contribution:** 3
**Rating:** 6
**Confidence:** 4

**Summary:**

This paper introduce a text-to-3D scene generation method called CompGS, where a compositional optimization strategy is proposed to improve the generation of multiple objects.

**Strengths:**

The paper is well written and easy to follow. The generated results are significantly better than existing work。

**Weaknesses:**

1. Object scale. This paper introduces a reasonable improvement by using rescaling during the entity-stage optimization. However, it does not address the scale relationship of the final generated objects. For instance, in Figure 1, the soccer ball and butterfly appear disproportionately large in the scene. I am curious if this is a bias introduced during training, as such issues do not typically arise in the inialization of text-to-image models. This could indicate a potential limitation in the current approach’s handling of object scaling.
2. Decomposing optimization: The decomposition approach does not consider the relationships between objects. The decompositional text embedding v_l may conflict with the actual 3D scene. For example, in Figure 4, the use of “a girl” as the text prompt to guide the SDS optimization generally results in a standing figure, which does not align with the curled posture of the girl in the scene. This discrepancy might impact the quality of the generated output, as evidenced by the slightly unnatural appearance of the girl in Figure 4.
3. Evaluation metrics: I think that CLIP can only evaluate the alignment between the generated output and the input text, but it does not adequately reflect the quality of the generation. The authors might consider incorporating aesthetic evaluation metrics, e.g., LAION Aesthetics Predictor [1] and PickScore [2], to provide a more comprehensive assessment of the aesthetic quality of the generated results.

[1] https://laion.ai/blog/laion-aesthetics. [2] Pick-a-Pic: An Open Dataset of User Preferences for Text-to-Image Generation.

**Questions:**

See the weaknesses section.

---

### Official Review · Reviewer_HYdB · 2024-11-06

**Soundness:** 4
**Presentation:** 4
**Contribution:** 2
**Rating:** 6
**Confidence:** 4

**Summary:**

This paper presents CompGS, a novel framework for compositional text-to-3D generation leveraging 3D Gaussian Splatting (GS) and Score Distillation Sampling (SDS) optimization. CompGS enables the generation of 3D content with multiple objects and complex interactions by decomposing and optimizing objects at both entity and composition levels, achieving enhanced detail in smaller objects through dynamic parameter adjustments.

Key contributions include transferring 2D compositionality from text-to-image models to initialize 3D Gaussian parameters, along with a dynamic optimization strategy based on Score Distillation Sampling (SDS) to maintain 3D consistency across objects.

CompGS demonstrates superior performance in compositional quality and semantic alignment compared to existing methods on the T3Bench dataset, also offering potential for controllable 3D editing.

**Strengths:**

1. The innovative use of 2D compositionality to inform 3D initialization allows for the efficient and consistent generation of multi-object 3D scenes.
2. Dynamic optimization based on Score Distillation Sampling enables effective control over both fine-grained and large-scale details in 3D objects.
3. Quantitative and qualitative evaluations on T3Bench indicate high compositional integrity and visual quality, marking an improvement over existing compositional 3D methods.
4. The paper is well-structured and easy to follow, with a clear description of the models employed.

**Weaknesses:**

1. The paper lacks experiments that test small objects generation, such as creating a scene with a castle and a figure standing on it, where the model captures fine details in smaller objects.
2. The paper does not address scenarios involving scenes with a high density of entities, such as a garden with hundreds of entities. While generating scenes with only a few entities can be done manually by positioning objects individually, testing whether the model can generate large-scale scenes that align with the text description would better demonstrate its robustness.

**Questions:**

1. Can CompGS generate large-scale scenes, such as a beach, a cityscape, or a cozy bedroom?
2. Why did the authors choose Score Distillation Sampling (SDS) as the loss function for optimization instead of Variational Score Distillation (VSD)? Is there empirical evidence that SDS produces better results than VSD in this setting?

---

### Note · Authors · 2024-11-14

**Comment:**

Dear Reviewers and Area Chairs,

Thank you for the time and effort invested in reviewing our paper. We appreciate the valuable feedback provided and have decided to withdraw our current submission to revise and enhance our paper based on your suggestions.

Best regards,

The Authors

**Withdrawal Confirmation:**

I have read and agree with the venue's withdrawal policy on behalf of myself and my co-authors.